# Enhancing performance with multisensory cues in a realistic target discrimination task

**Caterina Cinel** [1]*, **Jacobo Fernandez-Vargas**[1], **Christoph Tremmel**[1,2], **Luca Citi**[1], **Riccardo Poli**[1]

**1** Brain Computer Interface and Neural Engineering Lab, School of Computer Science and Electronic Engineering, University of Essex, Colchester, United Kingdom, **2** WellthLab, Electronics and Computer Science, University of Southampton, Southampton, United Kingdom

* ccinel@essex.ac.uk

## Abstract

Making decisions is an important aspect of people's lives. Decisions can be highly critical in nature, with mistakes possibly resulting in extremely adverse consequences. Yet, such decisions have often to be made within a very short period of time and with limited information. This can result in decreased accuracy and efficiency. In this paper, we explore the possibility of increasing speed and accuracy of users engaged in the discrimination of realistic targets presented for a very short time, in the presence of unimodal or bimodal cues. More specifically, we present results from an experiment where users were asked to discriminate between targets rapidly appearing in an indoor environment. Unimodal (auditory) or bimodal (audio-visual) cues could shortly precede the target stimulus, warning the users about its location. Our findings show that, when used to facilitate perceptual decision under time pressure, and in condition of limited information in real-world scenarios, spoken cues can be effective in boosting performance (accuracy, reaction times or both), and even more so when presented in bimodal form. However, we also found that cue timing plays a critical role and, if the cue-stimulus interval is too short, cues may offer no advantage. In a post-hoc analysis of our data, we also show that congruency between the response location and both the target location and the cues, can interfere with the speed and accuracy in the task. These effects should be taken in consideration, particularly when investigating performance in realistic tasks.

## Introduction

In everyday life we constantly make decisions. Some of those decisions are complex, requiring considerable amount of information and reflection, while others can be based on simple perceptual features, requiring for example rapid discrimination of external stimuli. In many cases, decisions can be highly critical in nature, with mistakes possibly resulting in extremely adverse consequences. Yet, often decisions have to be made quickly and with limited information. This is the case, for example, of situations where fast reactions to sudden stimuli are critical, for example, when driving or in a military context.

**Data Availability Statement:** All original data are deposited at the University of Essex Research Data Repository (https://researchdata.essex.ac.uk/), item ID. This is the DOI for the data repository https://dx.doi.org/10.5526/ERDR-00000144.

**Funding:** The authors acknowledge support of the UK Defence Science and Technology Laboratory (Dstl) and Engineering and Physical Sciences Research Council (EPSRC) under grant EP/P009204/1. This is part of the collaboration between US DOD, UK MOD and UK EPSRC under the Multi-disciplinary University Research Initiative. Dstl contributed to the early stages of the study design, and had no role in data collection and analysis, decision to publish, or preparation of the manuscript. EPSRC had no role in study design, data collection and analysis, decision to publish, or preparation of the manuscript. This article is an overview of UK MOD sponsored research and is released for informational purposes only. The contents of this article should not be interpreted as representing the views of the UK MOD, nor should it be assumed that they reflect any current or future UK MOD policy. The information contained in this article cannot supersede any statutory or contractual requirements or liabilities and is offered without prejudice or commitment.

**Competing interests:** The authors have declared that no competing interests exist.

This study was part of a US-UK collaboration in a Multidisciplinary University Research Initiatives (MURI) program, that aims at developing Brain-Computer Interfaces (BCIs) capable of enhancing decision-making performance in multisensory environments. In previous work, we have studied how, in those conditions, BCIs can improve accuracy of such decisions thereby acting as a form of cognitive enhancement [1–3]. In those studies, participants were presented with video streams of realistic scenarios (patrolling a dark corridor and manning an outpost at night) where users need to identify, as friends or foes, and as quickly and accurately as possible any unidentified uniformed soldiers that may present themselves [4, 5].

Within this context, given the adverse consequences associated with an incorrect or excessively slow identification, and given the ever increasing reliance on technology to assist soldiers, it seemed reasonable and ethical to attempt improve decision speed and accuracy by assisting the decision makers by providing them with additional information through verbal, auditory or audio-visual, communication from a remote human or AI assistant. To achieve this goal, key questions are, of course, what are the best timing and best perceptual modality (or modalities) to communicate the additional information, so that it can help rather than confuse, distract or delay the decision maker.

As a first step in this direction, in this study, we explore whether perceptual decisions in a patrolling task can be enhanced with the use of communication cues informing the decision maker of the approximate location (hemifield) in which an imminent target stimulus (a soldier) is about to appear. To help simplify the analysis, we assumed that the advice is always correct and we also considered a simplified version of the corridor scenarios, where two static images were presented, instead of video streams.

A preliminary analysis of this experiment was presented in a short conference paper in [6].

## Background

In this section we review relevant literature and explain its links to our study.

### Unimodal spatial cueing

In the literature, there is evidence showing that response times to a sudden visual stimulus can be accelerated when its spatial location is cued by a preceding stimulus (if cue and stimulus appear one after the other within a specific temporal window), compared to when no cue is presented. This has been extensively studied, particularly in simple tasks, using the classic *spatial cueing* paradigm [7–9]. Typically there is a differentiation between *exogenous cues* and *endogenous cues*. Exogenous cues are presented at the cued location and cause attention to automatically and *involuntarily* move to it before the task-relevant stimulus is presented. Endogenous cues are symbolic cues, such as an arrow pointing towards the direction where the stimulus will appear, or a word (e.g., "left" or "right"), providing spatial information without being in the cued location. These cause a *voluntary* and rapid shift of attention towards the referred location. Furthermore, in spatial cueing studies, typically a varying proportion of *invalid cues* (in which the target stimulus appears in the location that is *not* the one indicated by the cue, resulting in even worse performance than in the absence of a cue) are presented amongst valid cues.

When using simple stimuli in classic cueing paradigms, with exogenous cues, the advantage in validly cued trials is typically observed with a cue-stimulus interval between 300 and 500 ms. Intervals longer than that tend to produce the opposite effect, with valid trials resulting in slower reaction RTs compared to neutral trials—an effect known as *inhibition of return* (IOR; [10]). However, whether or not this is the case for endogenous cues is more controversial.

Spatial cueing has also been studied in realistic situations. For example, in driving simulations [11–13] or air traffic control simulations [14, 15] a spatial cue facilitates a prompter response to a potentially dangerous event, particularly in situations of high information load, where situation awareness can be particularly challenging. However, most of what is currently known of how spatial cueing can affect performance, and when and how it can be more effective, is the result of investigations carried out within a single modality—the visual perception domain.

## Cross-modal and multimodal spatial cueing

Research on crossmodal spatial attention has shown that spatial cueing works even when cue and stimulus are from different modalities [16–18] and, in more realistic conditions. For instance, Begault [19] found that the performance in a visual search task of airline crews improved when using spatial auditory cues in the same location of the target, compared to when the cue was a warning message ("traffic, traffic!") with no spatial information.

More recently Ho and Spence [12] conducted a series of experiments investigating the benefits of using auditory spatial cues in a simulated driving task. They examined the effects of non-spatial non-predictive, spatial non-predictive, and spatially-predictive sound cues, as well as symbolic predictive and spatially-presented symbolic predictive verbal cues. In all cases, auditory cues coming from the visual target relevant direction improved the ability to detect those targets (see also other investigations on cross-modal links in attention, e.g., [11, 14]). Moreover, the advantage was enhanced when non-spatial, but semantically informative cues (the words "front" and "back") were presented.

Consistently with Ho and Spence's findings [12], other studies [7, 12, 13, 20] have shown that spatial information cueing to the location or direction of a task-relevant visual stimulus can be provided not only by an exogenous cue but also by endogenous, meaningful directional cues—such as spoken words—presented in a non-relevant location (i.e., at the centre of the display, when the target-relevant visual stimulus is presented at left/right, top/bottom or front/rear locations).

## Multisensory perception

Evidence from studies on multisensory perception points to an advantage in terms of efficiency and accuracy for multisensory stimuli—as compared to unisensory stimuli—and in terms of enhanced perception [21–26]. This is has to be expected, given that in nature most external events are multisensorial. This is also confirmed in all instances in which there is a push toward forming coherent percepts when there is multisensory incongruency [27–31].

The multisensory advantage has been observed at behavioural level, whereby, for example, reaction times can be faster and accuracy greater for multisensory stimuli, compared to corresponding unisensory stimuli (e.g., [32, 33]). This also seems to apply to spatial cueing, whereby bimodal cues can be more effective in attracting attention to a location as compared to informative unimodal cues [17, 34].

At neural level, the multisensory advantage has also been observed both in form of multisensory neurons in several parts of animal brains, as well as in form of direct connections between unisensory areas [23, 35]. Our cognitive system seems to assume that multisensory events, that are very close temporally and spatially, belong together [27, 29, 36] and both behavioural and neural responses to the environment can be enhanced when the information provided is multisensory [25, 37–39].

## Temporal preparedness

In relation to the timing of stimulus presentation, it is well known that if a warning stimulus can be provided before a perceptual task (conveyed via a, so called, imperative stimulus) needs to be carried out, this may enhance the subject's preparation, resulting in shorter reaction times and possibly also higher accuracy [40–42]. Typically, when the temporal difference between warning and imperative stimulus (which is called the *foreperiod*) varies randomly, the longer the foreperiod, the shorter the RTs. In some conditions also accuracies vary with the foreperiod, although these are typically very high as imperative stimuli persist during the response period and, so, variations are relatively small.

## Implications for cueing in our target discrimination task

Most of the evidence reviewed above suggests that, when using cues to enhance performance in perceptual decision-making tasks, such as the one considered in our experiment, bimodal cues might be more effective than unisensory cues. However, this needed to be tested as there are several important differences between the conditions studied in the literature and our real-world setting. We consider them below.

In relation to the spatial cueing literature, in our experiment *we only have valid cues*, i.e., the information provided by the verbal communication is always correct. Naturally, this is only an approximation as, in the real world, one cannot always guarantee that the additional information provided to a decision maker will be 100% accurate. Also, *our stimuli were much more complex and realistic* than the simple stimuli used in classic cueing paradigms and, as we indicated above, *cues were endogenous*. For these reasons, it seemed unlikely that the short cue-stimulus intervals used in exogenous cues (300–500ms) would be maximally advantageous and, thus, we decided to explore longer cue-stimulus intervals (500, 700 and 900 ms). Because in our experiment participants knew that the target stimulus would always appear at the cued location, we did not expect to find IOR.

Also, while the cueing literature suggests that there is an advantage of providing cues and target stimuli in different sensory modalities, *it is not clear whether providing bimodal cues might further enhance performance*. For instance, would an audio-visual cue be more effective at moving attention to the location of an imminent visual stimulus as compared to an audio-only cue? This is particularly important in the context of our study.

Finally, the temporal preparation studies mentioned above suggest the possibility that temporal preparedness to the main task may also have an effect on performance (particularly, response times). In such research, *warning stimuli are typically uninformative* in relation to the imperative stimulus, while, of course, *spatial cues are informative*, particularly when they are always valid. Also, in the literature typically decision tasks associated with the imperative stimulus are much simpler (e.g., the stimulus persists during the decision, while in our study targets only appear for a very short time) and, so, accuracy is much higher (e.g., 98-99%) than those in our experiment (approximately 85%). Nonetheless, there is the potential for cues to influence the temporal preparedness, possibly contributing to improving performance. As we will discuss later, in some of the conditions of our experiment, cues are non-informative, so this allows us to explore to some degree whether there is a measurable foreperiod effect in our experiment.

## Additional potential effects in our experiment

In our experiment, the target stimulus was presented either on the right or left side of the screen, and participants were to respond using the left and right mouse button. Also, in part of the trials, a spoken cue saying the word "left" or "right" indicated where the stimulus was

going to appear. Therefore, a further aspect that needed to be examined was whether performance was also affected by: a) a possible interference resulting from stimuli location (left of right of the display) and the response position (left or right mouse click), known as the *Simon effect* [43, 44], and b) the interference that might originate between the response location and the spoken cue (e.g., "left" vs. "right"), which is known as the *spatial Stroop effect* [45]. In this case, a semantic interference may arise when the spoken cue-word is incongruent with the location of the left and right arrow keys or the left and right buttons of a mouse.

We should note that the Simon effect has been shown to be determined by the relative location of a stimulus, rather than absolute location in the visual field [45, 46].

Such effects are an important aspect that can have practical implications in realistic environments, where often tasks are based on spatial features and manual responses. We will study this in a post-hoc analysis.

## Materials and methods

In this study, we performed an experiment where participants had to rapidly decide whether a uniformed character, appearing suddenly and for a very short time, on the left or right side of a poorly-lit realistic corridor, was wearing a cap or a helmet (see Fig 1). The experiment was

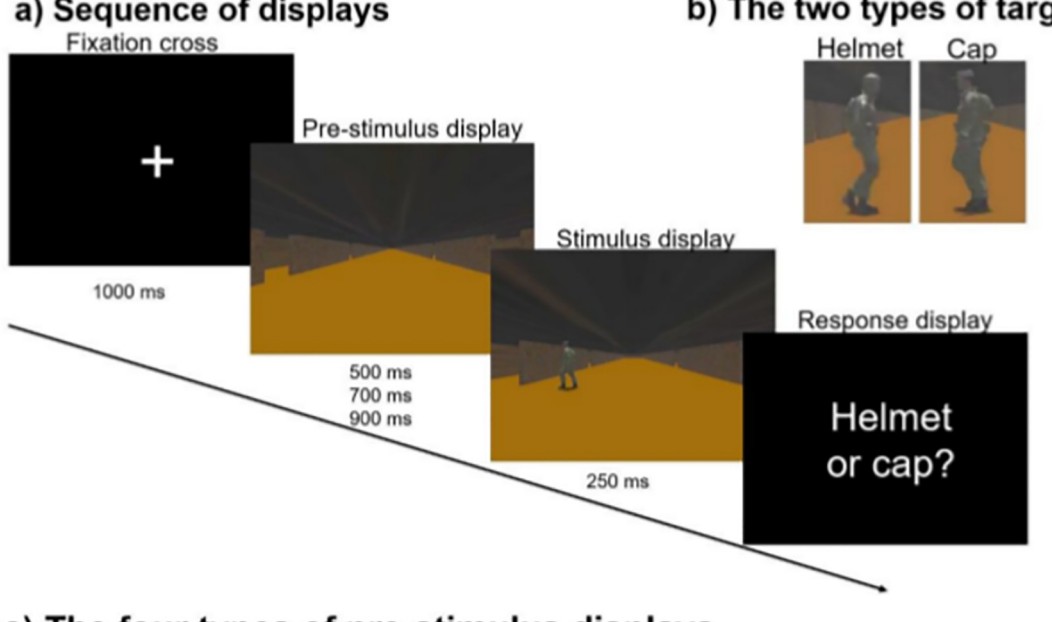

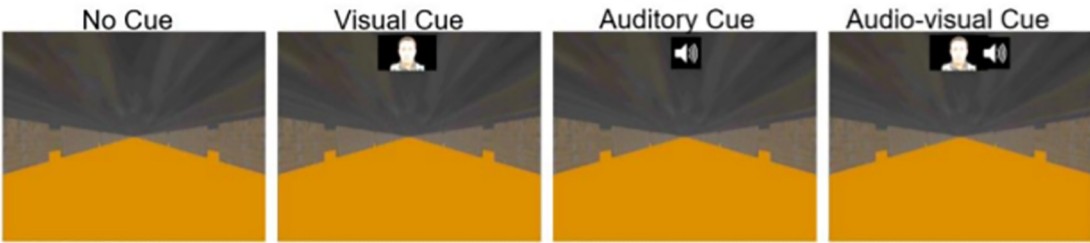

**Fig 1.** a) The sequence of displays on each trial and their duration. The SOA varied randomly between 500, 700 and 900 ms, while the response display lasted until a response was given; b) The two types of target characters appearing in the stimulus display (helmet or cap); c) In each block, one of the four types of pre-stimulus displays was shown: No Cue, Visual Cue, Auditory Cue or Audio-Visual Cue.

divided into blocks. In some of the blocks, the stimulus display was preceded by a cue indicating whether the character/soldier would appear on the left or the right. The cue was either as a spoken word ("left" or "right") only, or in the form of a synthetic face uttering the cue word (with matching lip movements). Performance was then measured in terms of both accuracy and reaction times. More details are provided below.

## Stimuli and procedure

Participants were seated at a distance of about 80 cm from the computer monitor, where the stimuli were presented. They were presented with displays (in full screen) showing an image of a long corridor with doors on either side, and where a uniformed character appeared (in a posture suggesting the character is walking across the corridor), for 250 ms (see Fig 1). Participants had to decide, as rapidly as possible, whether the character was wearing a helmet or a cap (see Fig 1(b)).

The corridor was always shown centrally on the display, so that the two side walls of the corridor were always symmetric with respect to the screen middle line (see Fig 1). The character could appear at different positions in the corridor, either horizontally (i.e., more or less lateral w.r.t. the centre of the display) or vertically (i.e., as if further away or nearer to the observer). The size of the character varied between 3.41˚ and 5.37˚ in height and 4.17˚ and 1.27˚ horizontally (reflecting the varying distances alongside the corridor from the observer). The horizontal distance of the character from the centre of the corridor varied between 1.54˚ (nearer to the centre of the corridor) and 16.22˚ (nearer to the left or right wall). So, it was never exactly at the centre.

Each trial started with a display showing a fixation cross for one second. This was followed by a *pre-stimulus display* representing the corridor. In some of the trials a *cue* was also presented together with the pre-stimulus display. When a cue was presented, this was either an *Auditory Cue* (AC) or an *Audio-Visual Cue* (AVC). The AC consisted of a voice uttering either the word "left" or the word "right", to indicate the side of the corridor where the character would appear (with 100% validity) in the following display. The spoken cue was played by a loudspeaker positioned centrally, behind the monitor where the visual stimuli were presented. The AVC consisted on a face, positioned at the top centre of the display (as shown in Fig 1(c)), uttering either the word cue "left" or the word "right", with sound; note that the lip movements of the face pronouncing the spoken cue was only an approximation of the correct lip movements, and, so, on their own they were insufficient to understand what word was being pronounced.

In addition, in some of the trials a *static face* was presented concurrently with the pre-stimulus display. The face was exactly the same as in the AVC condition, except that no lip movement nor sound was present. For simplicity, we called this *Visual Cue* (VC) (see Fig 1(c)), even though it did not act as a cue, i.e., it did not provide any information. We included VCs in the experiment to verify that the face used in AVCs, on its own, would not facilitate the task.

Finally, in *No Cue* (NC) trials (see Fig 1(c)), only the pre-stimulus display (empty corridor) was shown.

The experiment consisted of 16 blocks of 48 trials each. In each block, the type of pre-stimulus display was fixed, with four blocks for each pre-stimulus display type. The block order was randomised for each participant, and each block was preceded by a display informing the participant about the type of cue in that block. At the end of each block, the mean accuracy for that block was displayed as a form of feedback. *Reaction Times* (RTs) and *accuracy* were recorded.

The time interval between the pre-stimulus display and the target stimulus—also known as *Stimulus Onset Asynchrony* (SOA)—was varied randomly within each block between 500, 700 or 900 ms.

## Participants and ethical approval

Thirty-three participants in total took part in the experiment. However, data from participants whose accuracy in the task was below 60% (three participants) and participants who did not complete the experiment (two participants) were discarded. Therefore, here we present data from *28 participants* in total. All participants had normal or corrected-to-normal vision.

Participants signed a written informed consent form before taking part in the study. The research received ethical approval by the UK Ministry of Defence Research Ethics Committee and the University of Essex in June 2017 (Application Number: 832/MoDREC/17). All experiments were performed in accordance with the relevant guidelines and regulations.

## Rejection criteria and statistical analyses

Reaction times longer than 1.5 second were considered outliers and discarded (on average 4% of trials were removed). In addition, as is typical in cognitive experiments, incorrect responses were also discarded from RT analyses.

Data were analysed with within-subjects ANOVAs and paired t-tests, with the Benjamini-Hochberg correction for multiple tests. Effect sizes were measured using the partial $\eta^2$.

## Results

### Effects of auditory cues and audio-visual cues

In this section the following two hypotheses were tested: a) That responses in trials with the spoken cue (both AC and AVC trials) are faster and more accurate than responses in trials with no spoken cue (NC and VC trials), and b) that responses in AVC trials are faster and more accurate than responses in AC trials. We had no specific hypotheses regarding any differences between NC and VC trials, and no specific hypothesis regarding the effect of the SOA interval, which was investigated to find the optimal SOA interval for boosting performance in realistic environments.

**Reaction times.** Mean RTs, according to SOA and pre-stimulus type, are displayed in Fig 2. The mean RT across all conditions was 489 ms (SD 107 ms).

A 3 × 4 within-subjects ANOVA showed that there was a *main effect of SOA on RT* ($F_{(2, 27)}$ = 13.47, $p < .001$, partial $\eta^2$ = .33), indicating that *RTs were slower with shorter SOAs* (see "Reaction Times" column in Table 1).

Pairwise comparisons of all SOA pairs (regardless of pre-stimulus condition) showed that *RTs in the 500 ms SOA were the slowest* ($p < .001$ *vs* 700 ms SOA, and $p < .001$ *vs* 900 ms SOA), while the difference in RTs between the 700 ms and 900 ms SOAs was non-significant ($p = .473$).

The ANOVA showed that there was *no significant main effect of pre-stimulus type on RT* ($F_{(3, 27)}$ = 2.04, $p = .114$, partial $\eta^2$ = .07), and, accordingly, pairwise comparisons of all pre-stimulus conditions (see "Reaction Times" column in Table 2) confirmed no significant results (all $p > 0.1$).

There was, however, a *significant interaction between pre-stimulus and SOA* ($F_{(6, 27)}$ = 5.038, $p < .001$, partial $\eta^2$ = .16). We, therefore, performed pairwise comparisons (one-tail t-tests) of pre-stimulus within each SOA, to test whether: a) the spoken cue in AC and AVC trials would give a respective advantage over the NC and VC conditions (the spoken cue was

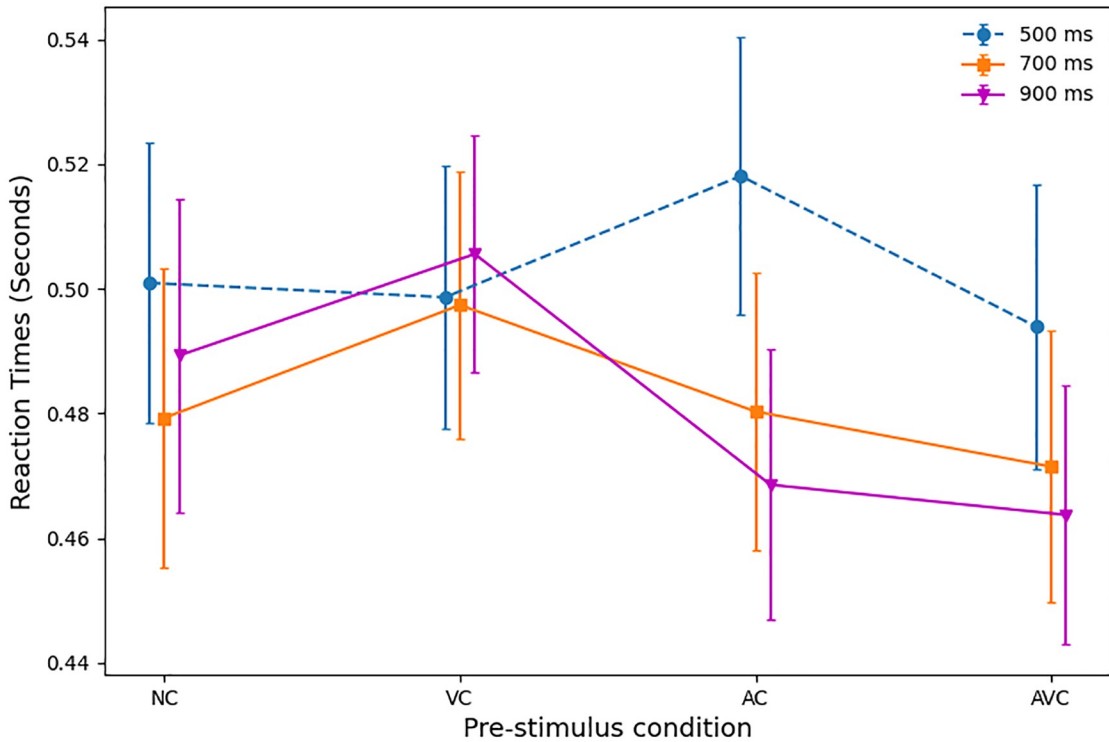

**Fig 2. Mean RTs with standard errors, according to SOA (500, 700 and 900 ms) and pre-stimulus type (NC, VC, AC and AVC).**

100% predictive); b) performance in AVC trials would be better than performance in AC trials. The corresponding *p*-values are shown in Table 3. This shows that there is a significant difference for the 500 ms SOA condition, where in AC trials responses were slower than in AVC trials. Statistically significant differences were also found within the 900 ms SOA. Here RTs in both AC and AVC trials were significantly faster than in VC trials, and RTs in AVC trials were also significantly faster in NC trials. All differences within the 700 ms SOA were not statistically significant.

Finally, we compared performance in each pre-stimulus condition between the different SOAs. As shown in Table 4, for RTs, in cued conditions, statistically significant differences exist for both the AC and AVC conditions between the 500 ms SOA and the two longer SOAs; additionally, RTs were significantly faster in AC trials with a 900 ms SOA than with a 700 ms

**Table 1.** Top half: RTs (in seconds) and accuracy (proportions of correct responses), with standard deviations, in the different SOA conditions, regardless of pre-stimulus type. Bottom half: p-values of pairwise comparisons (one-tail t-tests, Benjamini-Hochberg adjusted), with statistically significant values in bold ($\alpha$ = .05).

|  | Reaction Times | Accuracy |
|---|---|---|
| 500ms | 0.502 (0.109) | 0.855 (0.061) |
| 700ms | 0.482 (0.110) | 0.870 (0.052) |
| 900ms | 0.481 (0.106) | 0.872 (0.056) |
|  | **p-values** | |
| 500ms vs. 700ms | **<0.001** | **0.041** |
| 500ms vs. 900ms | **<0.001** | **0.017** |
| 700ms vs. 900ms | 0.473 | 0.315 |

**Table 2.** Top half: RTs (in seconds) and accuracy (proportions of correct responses), with standard deviations, in the different pre-stimulus conditions, regardless of SOA. Bottom half: p-values of pairwise comparisons (one-tail t-tests, Benjamini-Hochberg adjusted), with statistically significant values in bold ($\alpha$ = .05).

|  | Reaction Times | Accuracy |
|---|---|---|
| AC | 0.489 (0.112) | 0.871 (0.057) |
| AVC | 0.476 (0.111) | 0.885 (0.051) |
| NC | 0.490 (0.120) | 0.862 (0.069) |
| VC | 0.500 (0.104) | 0.846 (0.060) |
|  | p-values | |
| VC vs. NC | 0.892 | 0.955 |
| AC vs. NC | 0.535 | 0.157 |
| AC vs. VC | 0.165 | **0.007** |
| AVC vs. NC | 0.165 | **0.020** |
| AVC vs. VC | 0.165 | **0.007** |
| AVC vs. AC | 0.165 | **0.044** |

SOA. On the contrary, in the two non-cued/control conditions, NC and VC, only one statistical difference was identified: namely, for NC, between the 500 ms and 700 ms SOAs.

We should note that RTs in the NC and VC conditions did not show the typical slope associated with the foreperiod effect. These two conditions are the most similar to those used in foreperiod studies as the corresponding pre-stimulus displays are non-informative and so can be thought of as warning stimuli. The absence of statistical evidence for a foreperiod effect may be due to additional resources required to parse the NC and VC displays (these displays have a much greater complexity than a typical warning stimuli) resulting in a reduced preparation for the imperative stimulus irrespective of SOA, or to the effect being relatively small compared with other effects.

**Accuracy.** The overall mean accuracy (across all participants and conditions) was 86.6% (SD = 5.4%). However, accuracy varied depending on SOA and pre-stimulus, as shown in Fig 3.

**Table 3. Benjamini-Hochberg adjusted p-values from paired one-sided t-tests for pre-stimulus differences within each SOA, for reaction times (RTs) and accuracy ($\alpha$ = 0.05).** Statistically significant values are in bold.

| Reaction Times | 500ms | 700ms | 900ms |
|---|---|---|---|
| VC < NC | 0.606 | 0.930 | 0.894 |
| AC < NC | 0.951 | 0.653 | 0.086 |
| AC < VC | 0.951 | 0.164 | **0.010** |
| AVC < NC | 0.606 | 0.418 | **0.043** |
| AVC < VC | 0.606 | 0.164 | **0.010** |
| AVC < AC | **0.025** | 0.418 | 0.375 |
| **Accuracy** | **500ms** | **700ms** | **900ms** |
| VC > NC | 0.911 | 0.970 | 0.899 |
| AC > NC | 0.600 | 0.348 | 0.151 |
| AC > VC | 0.129 | **0.024** | **0.012** |
| AVC > NC | 0.255 | **0.041** | 0.899 |
| AVC > VC | 0.091 | **<0.001** | 0.720 |
| AVC > AC | 0.208 | 0.053 | 0.989 |

**Table 4. Benjamini-Hochberg adjusted p-values from paired one-sided t-tests for differences between SOAs (all pairings) within each cue type, for reaction times (RTs) and accuracy ($\alpha$ = 0.05).** Statistically significant values are in bold.

| RTs | NC | VC | AC | AVC |
|---|---|---|---|---|
| 500 > 700 | **0.047** | 0.852 | **<0.001** | **0.011** |
| 500 > 900 | 0.222 | 0.852 | **<0.001** | **<0.001** |
| 700 > 900 | 0.838 | 0.852 | **0.048** | 0.180 |
| **Accuracy** | **NC** | **VC** | **AC** | **AVC** |
| 500 < 700 | 0.469 | 0.298 | 0.068 | 0.063 |
| 500 < 900 | 0.469 | 0.294 | **0.033** | 0.063 |
| 700 < 900 | 0.678 | 0.298 | 0.129 | 0.643 |

A 3 × 4 within-subject ANOVA showed that there was a significant main effect of SOAs (F(2,27) = 4.43, p = .017, partial $\eta^2$ = .14), with accuracy being lower with the 500 ms SOA (85.6%, SD = 6.1), followed by the 700 ms (87%, SD = 5.2) and higher with the 900 ms SOA (87.2%, SD = 5.6). We tested all pairings of the three SOAs, which showed a significant difference between the 500 ms and 700 ms SOAs (p = .041), and between the 500 ms and 900 ms SOAs (p = .017), but not between 700 ms and 900 ms SOAs (p = .315) (see Table 1).

The ANOVA also showed a significant main effect of pre-stimulus (F(3,27) = 7.97, p <.001, partial $\eta^2$ = .23), with AVC trials being the most accurate, followed by AC trials, NC trials and VC trials (see Table 2). Pairwise comparisons of the pre-stimulus variable, independently of SOA, showed that only the difference between AC and NC trials was non-significant

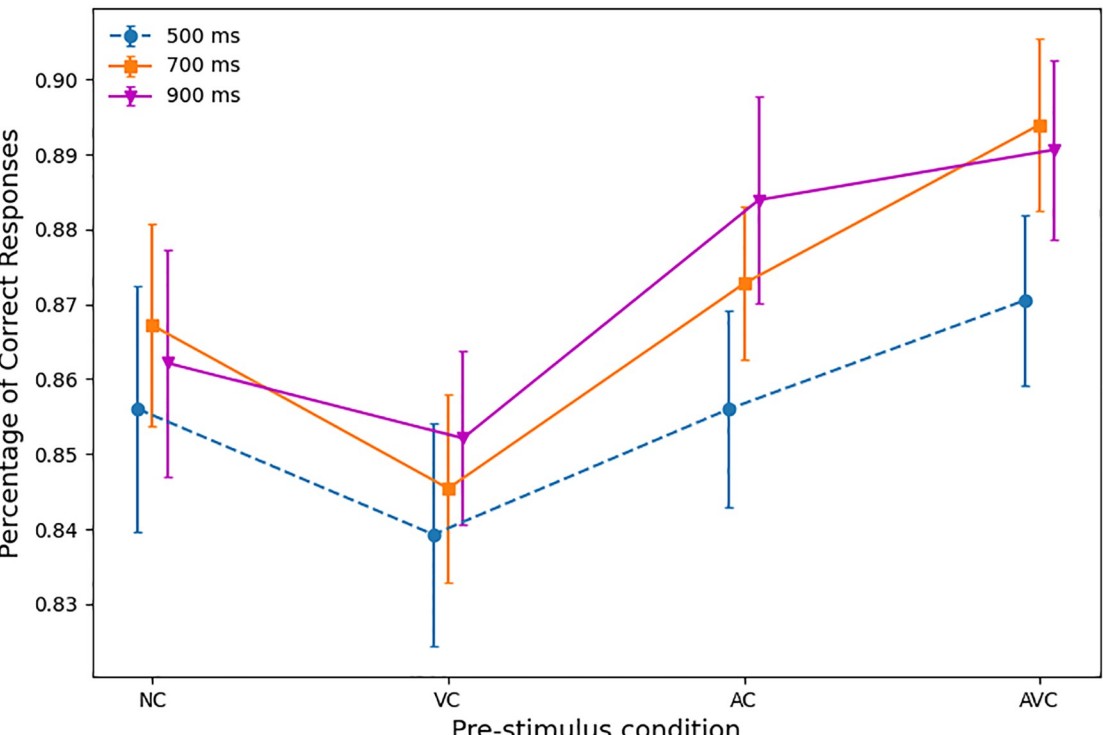

**Fig 3. Mean accuracy (percentages) with standard error according to SOA (500, 700 and 900 ms) and pre-stimulus (NC, VC, AC and AVC).**

(p = .131), while all differences between AC and VC (p <.001), AVC and NC (p = .017), AVC and VC (p <.001), AVC and AC (p = .037) were significant (see Table 2).

The interaction between the pre-stimulus and SOA was non-significant (F <1, partial $\eta^2$ = .02). In pairwise comparisons, where differences between pre-stimuli were tested separately for the three SOAs, the results were the following (see also Table 3). In the 500 ms SOA, all differences were non-significant (all p-values > 0.7). In the 700 ms SOA, accuracy was statistically greater in AC than VC trials (p = .029), while for AVC trials, accuracy was statistically greater than for NC trials (p = .034), VC trials (p = .0002) and AC trials (p = .044). Finally, with the 900 ms SOA the only significant difference was between AC and VC trials (p = .01).

Finally, if we look at the differences in accuracy for different SOAs and pre-stimulus conditions in Table 4, we see that none of the non-cued (NC and VC) comparisons is statistically significant, while one comparison (AC 500 < 900) is statistically and three more are nearly statistically significant in the cued conditions, effectively confirming the corresponding RT results.

**Overall performance.**   In general, the results presented above indicate that there is a beneficial effect in the cued (AV and AVC) *vs* non-cued (NC and VC) conditions, which is typically manifested in either RTs (SOA of 900 ms) or accuracy (at the 700 ms SOA). However, a joint improvement of RT and accuracy is present only in one case (see AC *vs* VC comparisons in Table 3).

Therefore, formally, our first hypothesis, i.e., that responses in trials with the spoken cue would be faster *and* more accurate than responses in trials with no spoken cue, was not fully confirmed. However, what was confirmed is that, in most cases, *the performance in trials with the spoken cue is better than in trials without it*. Both accuracy and response time are important when making critical, time-sensitive decisions. While, ideally, one would want to improve both, the individual improvements seen in our experiment may still be quite valuable.

We should note that this performance improvement is unlikely to be simply due to a foreperiod effect, for two reasons. Firstly, as we reported above, if a foreperiod was prominent in our experiment, it would have manifested itself first and foremost in the non-cued conditions (NC and VC), which is not the case. Secondly, in AC and AVC trials—unlike in NC and VC trials—the appearance of the pre-stimulus display is not a simple warning signal, as the cue is informative. At the presentation of the cue the participant needs to do three tasks: (1) interpretation of the auditory cue, (2) orienting the attention/gaze to the correct hemifield, and (3) temporal preparation to the main task (the main task being determining if the uniformed character is wearing a helmet or a cap). Only task (3) could contribute to what is normally described as the foreperiod effect and, yet, there seems to be no such an effect in then NC and VC conditions where only task (3) needs to be carried out. Therefore, the faster RTs with longer SOAs in AC and AVC are more likely a consequence of the fact that longer SOAs are needed for the interpretation of the endogenous cue.

Interestingly, a bimodal effect, where the RTs for AVC were significantly faster than those for AV, was observed only at 500 ms SOAs (see Table 3). So, formally, our second hypothesis, i.e., that responses in AVC trials were faster and more accurate than responses in AC trials, was not fully confirmed. Upon reflection, in line with the literature on multisensory integration, one should not expect multisensory perception to provide advantages over unisensory perception if there is a low-level of noise (so, no need, for error correction) and enough time for making sense of the percepts is available. In our case, the spoken words (and also the moving lips) were noiseless. So, the most likely explanation for the superiority of AVC over AC only at the 500 ms SOA is the following: a 500 ms SOA does not always give enough time to process and react to the auditory cue in AC trials. So, the cue acts more like a distractor, resulting in the worst mean RT of all 12 pre-stimulus display and SOA pairs. In these more difficult

conditions, seeing also lip movements (in the AVC condition) provides sufficient advantages to bring back RTs to the levels seen in the no-cue conditions. In more real-world conditions, where auditory cues would be much noisier (e.g., being conveyed via noisy radio communication to a soldier in a noisy environment), it is likely that bimodal perception would have helped also at longer SOAs.

The combined effect of SOA and pre-stimulus on performance is well illustrated in Fig 4. This clearly shows not only that the audio-visual cues provide an advantage in both RTs and accuracy, but also that this effect is further enhanced by the longer SOAs. The criticality of the choice of timing is also illustrated by the AC at 500 ms SOA, in which a dramatic increase of the RTs is the result of the presentation of the cue too close to the target stimulus. The VC conditions were the least accurate and produced the slowest RTs. This is not surprising, as the visual cue alone does not give any additional information, unlike the AC and AVC trials, and it might actually act as a distractor, compared to the NC conditions.

## Congruency effects in our experiment

As explained in the Background section, it is possible that in this experiment there was interference when the response location (left and right mouse click) was incongruent with the target location (in all trials) and/or the spoken cue (in AC and AVC trials). This, in turn, could possibly modulate the facilitatory effect of the bimodal cue. If such an effect was present, we would expect comparatively slower RTs and/or lower accuracy when stimuli and response location were incongruent than when they were congruent [45].

In the experiment, the Simon effect might be observed on its own (in NC and VC trials), or in combination with the spatial Stroop effect (in AC and AVC trials), as the spoken cue was 100% predictive of the stimulus location. On the contrary, any spatial Stroop effect cannot be separated from the Simon effect.

An example of the two types of effects is illustrated in Fig 5 for an AVC trial.

Note that in our experiments the stimulus location is irrelevant to the task, as the task is to decide whether the character was wearing a cap (right click) or a helmet (left click) irrespective of where it was located on the screen.

In the following subsections, we report the results of a series of *post-hoc analyses* aimed at elucidating to what degree the Simon and spatial Stroop effects could modulate performance and the potential benefits of bimodal cues in our experiment.

**Congruency between stimulus location and response location.**   In order to separate the effect of the character location from the effect of the spoken cue word (in the AC and AVC conditions), we first performed a statistical analysis where only NC and VC trials were included. As there is no effect of the spoken word, in NC and VC trials the only interference with the response location would originate from the location of the stimulus (Simon effect). Note that, we were not specifically interested in differences between NC and VC trials, but rather in differences between congruent and incongruent trials. A separate analysis for AC and AVC trials is presented in the next subsection.

Mean RTs for NC and VC for different SOAs are shown on the left-hand side of Fig 6, for congruent and incongruent trials. Overall, RT averages for congruent and incongruent trials were quite similar: 494 and 497 ms, respectively. Indeed, a $2 \times 3 \times 2$ ANOVA where the factors where Simon congruency (congruent and incongruent), SOA (500, 700 and 900 ms) and pre-stimulus type (NC and VC), showed that none of the main effects or interactions were significant (all $p > 0.1$).

However, when analysing *accuracy* for congruent and incongruent trials for NC and VC (see left-hand side of Fig 7), a $2 \times 3 \times 2$ ANOVA showed that there was a significant main effect

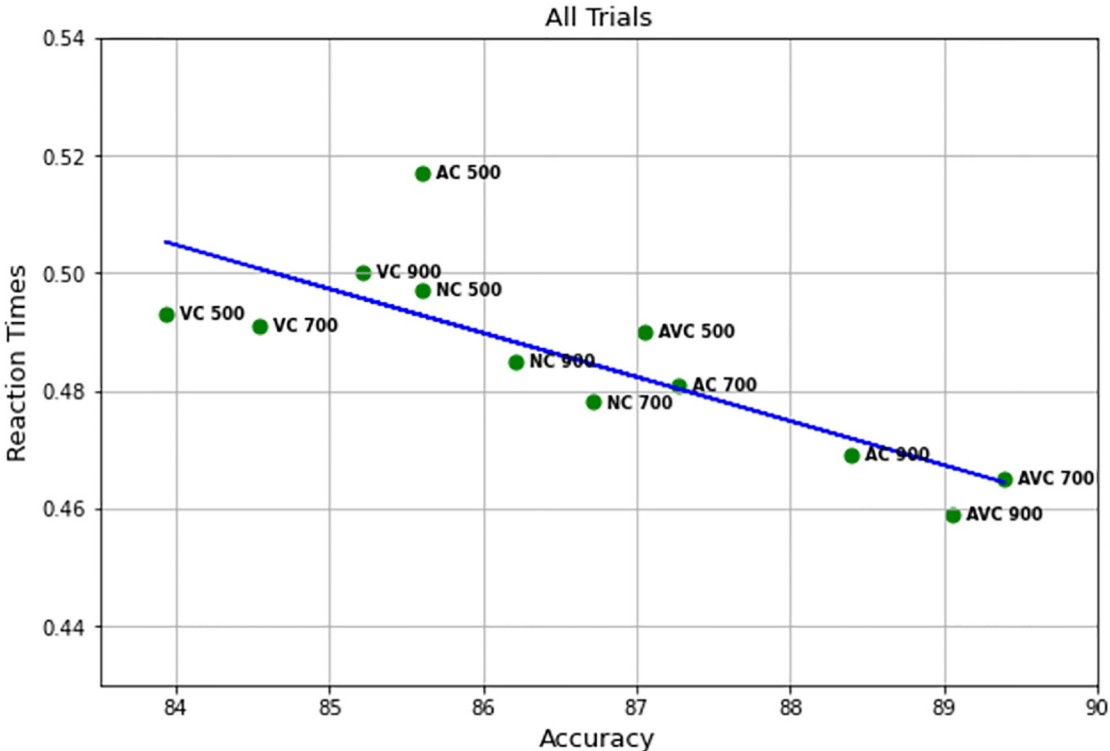

**Fig 4. Accuracy in relation to RTs for each of the twelve different conditions.**

of Simon congruency (F(1,27) = 6.227, $p$ = .019, partial $\eta^2$ = .18), where accuracy was 87% in congruent trials and 84% in incongruent trials. This is well illustrated in Fig 7, where all the blue markers (congruent trials) are distinctively separated from all red ones (incongruent trials). All the other main effects and interactions were non-significant (all $p$ > 0.1).

The results of these analyses suggest that, while the Simon effect did not affect RTs, there was an effect on accuracy. The effect is clearly illustrated on the left of Fig 8, where most markers for *congruent* NC and VC trials are shifted-right (better accuracy) versions of the corresponding markers for incongruent trials.

The lack of modulation in the response times is a partial inconsistency with the literature on the Simon effect (where differences in both accuracy and RTs are normally reported). This may be explained by the fact that, although the target was always in either the left or the right hemifield, the eccentricity of the target within the hemifield would vary, with some targets

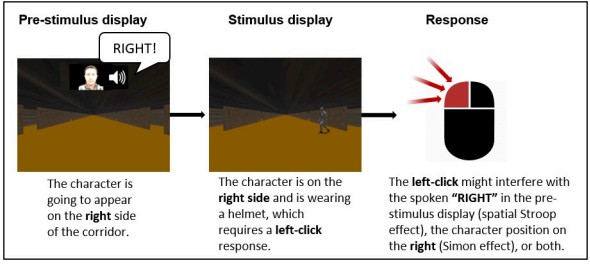

**Fig 5. An example of Spatial Stroop and Simon effects in an "incongruent" AVC trial.**

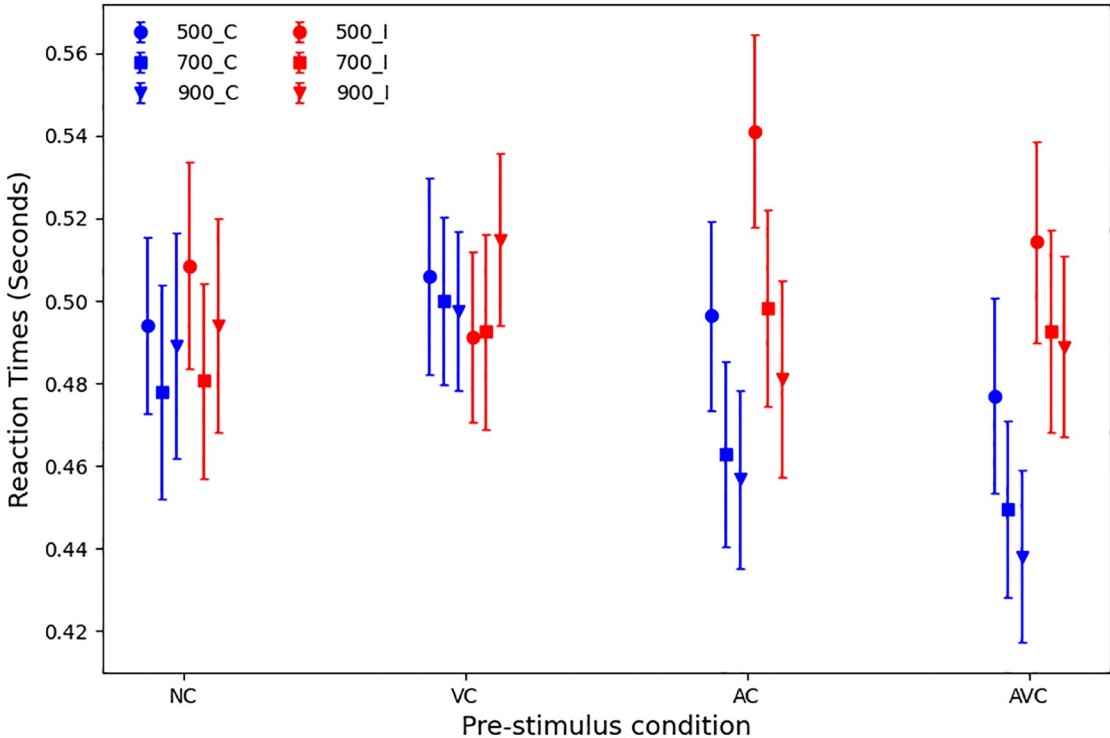

**Fig 6. Mean RTs with standard errors in congruent trials (blue) and incongruent trials (red), according to pre-stimulus display and SOA.**

being very close to the center of the corridor (thereby possibly resulting in virtually no interference), while others more clearly on one side. It is possible that this variable eccentricity results in performance differences between congruent and incongruent trials appearing only in the accuracy and not the RTs.

**Congruency between cue word and response location.** In AC and AVC trials (where there was the spoken cue), there is another possible source of interference, affecting the RTs and accuracy: the incongruence between the spoken cue-word and the response location (spatial Stroop effect).

To test the spatial Stroop effect, $2 \times 3 \times 2$ ANOVAs on both RTs and accuracy were performed, for the AC and AVC trials, with congruency, SOA and pre-stimulus type as the three factors.

The ANOVA analysis of the RTs (see also right hand side of Fig 6) showed that there was a significant main effect of Congruency (F(1,27) = 32.795, p <.001, partial $\eta^2$ = .55), with RTs being faster in the congruent condition than in the incongruent condition (463 and 503 ms, respectively). There was also a main effect of SOA (F(2,27) = 26.852, p <.001, partial $\eta^2$ = .50), 700 ms SOA, and RTs slower with the 700 ms SOA than with the 900 ms SOA (the mean RTs being 507, 476 and 466 ms, for the 500, 700 and 900 ms SOAs, respectively), which confirmed the finding from previous analyses. The main effect of pre-stimulus and all the interactions were non-significant (all $p$ >.1).

Similar results were obtained in an ANOVA analysis of the *accuracy* (see also right hand side of Fig 7), with the main effects all significant: (a) Congruency (F(1,27) = 16.018, p <.001, partial $\eta^2$ = .37), with higher accuracy in the congruent condition than in the incongruent condition (89.9% and 85.8%, respectively); (b) SOA (F(2,27) = 6.003, p = .004, partial $\eta^2$ = .18)

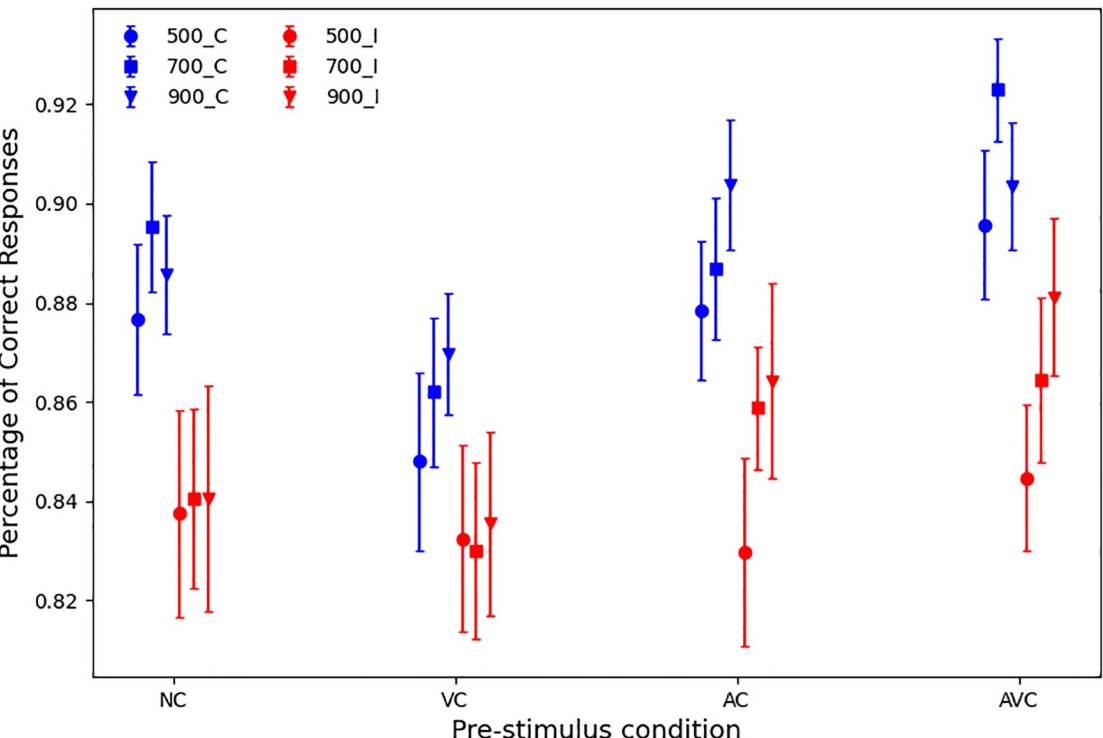

**Fig 7. Mean accuracy with standard errors in congruent trials (blue) and incongruent trials (red), according to pre-stimulus display and SOA.**

with accuracies 86.3%, 88.4% and 88.9%, for 500 ms, 700 ms and 900 ms SOAs, respectively; and pre-stimulus (F(1,27) = 4.407, p = .045, partial $\eta^2$ = .14), with accuracy being greater in AVC trials (89%) than in AC trials (87%). Pairwise comparisons for SOA showed a statistically significant difference between 500 and 700 ms ($p$ = .013), and between 500 and 900 ms ($p$ = .008). All interactions were non-significant (all $p >$.3).

Therefore, there was strong evidence of a congruency effect between the spoken word and the response location on performance. The effect is clearly illustrated in the middle and right of Fig 8, where AC and AVC markers for congruent trials are effectively shifted-right (better accuracy) and down (faster RTs) versions of the corresponding markers for incongruent trials. However, as there was no significant interaction between congruency and pre-stimulus, we can conclude that the beneficial effect of the bimodal cue did not depend on whether or not there was congruency between the spatial features of cue and stimulus and the response location.

## Discussion and conclusions

The main purpose of the study was to investigate the effectiveness of unimodal and bimodal cues at improving performance in a realistic decision-making task. In our experiment, the cues were presented to indicate the left or right location of the imminent target stimulus, and participants had to decide whether the target was wearing a helmet or a cap. The interval between cue and target (or SOA) could vary between 500 ms, 700 ms or 900 ms. Based on previous cuing research, we hypothesised that the auditory cue would give an advantage in the discrimination task, in terms of both RTs and accuracy. Additionally, based on previous multisensory literature showing that multisensory stimuli are in many cases processed faster and

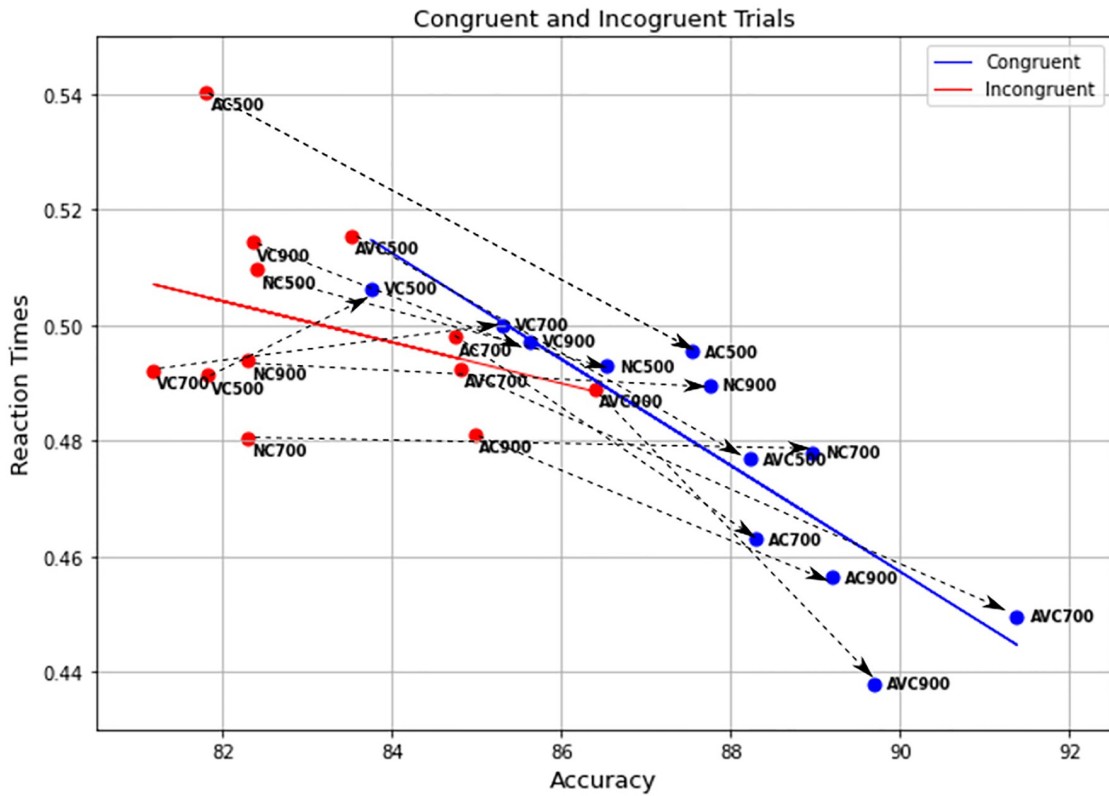

**Fig 8. Accuracy in relation to RTs for each of the twelve different conditions, for congruent (blue) and incongruent (red) trials.**
Dashed arrows connect corresponding conditions in congruent and incongruent trials.

more accurately than unisensory stimuli, we expected that these auditory cue advantages would be enhanced when this was combined with a "visual cue", which was in fact the face of a person uttering the cue words, resulting in a bimodal audio-visual cue.

Our data confirmed that, in general, there was an advantage in providing a cue. However, unimodal (auditory) cues were only beneficial with an SOA of 900 ms for RTs and an SOA of 700 ms for accuracy. In fact, using an SOA of 500 ms with unimodal cues resulted in the worst mean RT of all pre-stimulus display and SOA pairs. We argued that this is because a 500 ms SOA does not give enough time to process and react to the auditory cue and, so, the cue can act like a distractor. We found, however, that bimodal cues (in the AVC condition) were more effective than unimodal cues for RTs at precisely this SOA (500 ms), for both RTs and accuracy, effectively bringing back the RTs to the levels seen in the no-cue conditions. This suggests that seeing lip movements provides a multisensory advantage which is sufficient to compensate the distracting effect of an incompletely processed auditory cue. This, of course, can only be confirmed by further investigations.

Therefore, our two initial hypotheses, i.e., (1) that responses in trials with the spoken cue (both AC and AVC trials) are faster and more accurate than responses in trials with no spoken cue, and that (2) responses trials with a bimodal cue are faster and more accurate than responses in AC trials, were only partially confirmed.

We did not have any specific hypothesis in relation to the cue-stimulus intervals, as there was no clear indication from previous research of what the ideal temporal window should be to maximise the effect of the cues. As discussed above, we found that RTs tended to be faster

with longer intervals, particularly in cued trials, probably because 500 ms were not enough to process the spoken cʿue.

Our data also revealed a combined effect of SOA and pre-stimulus, as shown in Fig 8, showing a trend where the audio-visual cues provide an advantage in both RT and accuracy over audio only cue and no cue, which is further enhanced with longer SOAs. It is important to note that the face lip movements in bimodal cues were not understandable on their own (without the sound); however, they were slightly different for the two words and, therefore, after enough practice, participants could have implicitly learned to discriminate between the two, which might then have helped in the bimodal cues. And indeed a limitation of this study was the lack of a condition that could examine to what extent the lip movements helped to process the cue. Alternatively, another explanation as to how the bimodal effect might occur (and similar positive effect of multisensory stimuli reported in the literature), include the multisensory stimuli causing a "perceptual enhancement" [33].

In terms of practical lessons learnt from our study, in real-world scenarios, the person uttering the cue would not typically know how long after it the target would become visible. So, one could not choose beforehand the best modality of presentation, or, in fact, whether to withhold the cue to avoid distraction from/interference with the primary task. In this situation, our data show that one should always use bimodal cues, as they have proven to be the most resilient, never being worse, and in most cases being better (in terms of response speed, accuracy or both), than no-cue conditions. Naturally, future investigations will need to explore this aspect in more depth, and in particular, whether in more difficult conditions (e.g., in a noisy environment) bimodal cues can be even more effective at improving performance.

Finally, in our study we performed post-hoc analyses to examine whether there were any interference from the Simon effect and the spatial Stroop effect. In particular, we wanted to clarify whether the inconsistent effect of bimodal cues on the response times and accuracy could be explained by an interference from the incongruency between the response location (in the Simon effect) and the target stimulus location or the spoken word (in the spatial Stroop effect), or both. We did not find evidence that the Simon and Stroop effect limited the effect of bimodal cues, and there was no evidence of a significant Simon effect on reaction times, but there was a considerable effect on accuracy. Moreover, we found that there was a significant spatial Stroop effect on both RTs and accuracy, whereby responses were faster and more accurate when the spoken cue ("left" or "right") was congruent with the response location (left click or right click). This is an important aspect and it needs to be taken into account as it can have practical implications in real-life tasks, which are often based on spatial features and manual responses.

To conclude, our findings support the idea that, when used to facilitate perceptual decision under time pressure, and in condition of limited information in real-world scenarios, spoken cues can be effective in boosting performance (accuracy, reaction times or both), and even more so when presented in bimodal form. However, as seen in our scenario, cue timing plays a critical role and, if SOAs are too short (cf. 500 ms) cues may offer no advantage. While this work didn't have the specific objective to elucidate the precise mechanisms behind any performance improvements provided by unimodal or bimodal cues in our experiment, it suggested a number of avenues for both real-world applications and theoretical investigation. For example, a new experiment should include a control condition with moving lips, but without spoken cue, and further research is needed to clarify to what degree noisy conditions change the effect of unimodal and bimodal cues. More in general, further experiments are needed to investigate whether the advantage of presenting multisensory cues extends to other contexts and different types of stimuli and non-spatial cues. Finally, more research is needed on the influence of the

Simon and spatial Stroup effects in real-world situations and how these can be reduced using alternative interaction modalities.

## Acknowledgments

**Disclaimer**: We thank the editor and an anonymous reviewer for their constructive feedback.

## Author Contributions

**Conceptualization:** Caterina Cinel, Jacobo Fernandez-Vargas, Luca Citi, Riccardo Poli.

**Data curation:** Caterina Cinel, Jacobo Fernandez-Vargas, Christoph Tremmel, Riccardo Poli.

**Formal analysis:** Caterina Cinel, Jacobo Fernandez-Vargas, Christoph Tremmel, Luca Citi, Riccardo Poli.

**Funding acquisition:** Luca Citi, Riccardo Poli.

**Investigation:** Caterina Cinel, Jacobo Fernandez-Vargas, Luca Citi, Riccardo Poli.

**Methodology:** Caterina Cinel, Jacobo Fernandez-Vargas, Riccardo Poli.

**Project administration:** Luca Citi, Riccardo Poli.

**Resources:** Riccardo Poli.

**Software:** Caterina Cinel, Jacobo Fernandez-Vargas, Riccardo Poli.

**Supervision:** Luca Citi, Riccardo Poli.

**Validation:** Caterina Cinel, Jacobo Fernandez-Vargas.

**Visualization:** Caterina Cinel.

**Writing – original draft:** Caterina Cinel, Jacobo Fernandez-Vargas, Christoph Tremmel, Luca Citi, Riccardo Poli.

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
