## [Decision Letter · Decision Letter 0]

18 Jan 2022

PONE-D-21-38703Enhancing performance with multisensory cues in a realistic target discrimination taskPLOS ONE

Dear Dr. Cinel,

Thank you for submitting your manuscript to PLOS ONE. After careful consideration, we feel that it has merit but does not fully meet PLOS ONE’s publication criteria as it currently stands. Therefore, we invite you to submit a revised version of the manuscript that addresses the points raised during the review process. I found one expert reviewer to comment on your work. In order to ensure a timely review process and to give you a more precise guidance of how to potentially improve the manuscript, I decided to take the role of reviewing myself. Detailed suggestions can be found at the bottom of this letter. The referee considers your work important overall, yet there are some methodological issues that clearly prevent publication in its present form. This concerns the organisation of the manuscript, in particular the delineation of hypotheses, and the way the information is presented in the method section. I would invite you preparing a revision of your work that addresses all concerns together with a cover letter that provides point-by-point replies. 

We look forward to receiving your revised manuscript.

Kind regards,

Michael B. Steinborn, PhD

Academic Editor

PLOS ONE

Journal Requirements:

2.Thank you for stating the following in the Acknowledgments Section of your manuscript: 

[ This article is an overview of UK MOD sponsored research and is released for informational purposes only. The contents of this article should not be interpreted 

as representing the views of the UK MOD, nor should it be assumed that they reflect any current or future UK MOD policy. The information contained in this article cannot 

supersede any statutory or contractual requirements or liabilities and is offered without 

prejudice or commitment. The authors acknowledge support of the UK Defence Science and Technology Laboratory (Dstl) and Engineering and Physical Research Council (EPSRC) under grant EP/P009204/1. This is part of the collaboration between US DOD, UK MOD and UK EPSRC under the Multidisciplinary University Research Initiative.]

 [The work was funded by UK Defence Science and Technology Laboratory (Dstl) and Engineering and Physical Research Council (EPSRC) under grant EP/P009204/1

RP and LC received the award. https://epsrc.ukri.org/

The funders had no role in study design, data collection and analysis, decision to publish, or preparation of the manuscript.]

4.  We noticed you have some minor occurrence of overlapping text with the following previous publication(s), which needs to be addressed:

- https://ieeexplore.ieee.org/document/9215988

In your revision ensure you cite all your sources (including your own works), and quote or rephrase any duplicated text outside the methods section. Further consideration is dependent on these concerns being addressed.

**Editor Comments**

(-1-) theory and hypotheses 

I would completely agree with R1 that your theorising on the effects of multisensory cues is a bit confusing and so are the predictions derived from your theorising. I would suggest specifying what underlying process is assumed in your situation and what process model is assumed (or could be adopted) to predict performance effects on reaction time and accuracy. At present, you are debating general processes related to multisensory processing but it seems to fit not exactly to what your are examining empirically.   (-2-) downward-sloping SOA function  More specifically, it seems you are predicting a downward-sloping foreperiod function (the variable foreperiod effect, see doi:10.1037/xhp0000561; doi:10.1016/j.actpsy.2008.08.005), which you attribute to mechanisms of inhibition of return, which is incorrect in the present situation. I suggest reconsidering your theorising in the revised version of the manuscript.  (-3-) inhibition of return  The theorising on inhibition of return (IOR) is incorrect throughout the manuscript. Note that IOR is not an effect related to long intervals per se. Inhibition of return refers to the the way organisms gather information from visual systems by looking. For example, in a saccadic walk during the perception of a visual scene, the information processing system tends "not" to jump back to a previous saccadic point because it would prevent the system from gathering the information efficiently. It makes no sense to confuse the foreperiod effect with IOR.

Reviewers' comments:

Reviewer's Responses to Questions

**Comments to the Author**

1. Is the manuscript technically sound, and do the data support the conclusions?

Reviewer #1: No

2. Has the statistical analysis been performed appropriately and rigorously? 

Reviewer #1: Yes

3. Have the authors made all data underlying the findings in their manuscript fully available?

Reviewer #1: No

4. Is the manuscript presented in an intelligible fashion and written in standard English?

Reviewer #1: Yes

5. Review Comments to the Author

Reviewer #1: This study investigated how the efficiency of target discrimination changes depending on whether the pre-cue stimulus, which indicates the location of target presentation, was multisensory or unisensory. The main result is that multisensory cueing can be more beneficial for target discrimination than unisensory cueing. The authors concluded that processing more than one modality increases the benefit of the cue by some processes.

The paper addressed an interesting topic, and the results have the potential to be worthwhile to publish. However, I have some points that should be addressed prior to publication. I list these in my detailed review below. Particular care should be given to point 1, which concerns that it is not clear how the experimental situation in this study can be considered in the context of the previous studies, and this could pose problems in the overall interpretation of the results. Point 7 may help address this point.

1. The explanation in 61-82 needs to be more clarified in light of the authors' experimental situation. How does multisensory cueing increase the effectiveness of cues in directing spatial attention? The authors cited some citations, such as [37], that show multisensory cues guide attention better than unisensory cues. From this, the authors seem to hypothesize that multisensory cues work better than unisensory cues in their experiments. However, the literature cited here may have examined the effects of combining unisensory cues that function on their own. If so, then there is a problem with the connection between the previous studies and this hypothesis. The reason is that the visual cue in this study did not work by itself, as this is mentioned after the results section. Additionally, the auditory cue was not very effective on its own. This point is important because it can critically affect the reason for conducting each analysis and the overall interpretation of the results. Is there a different logic from the one I described?

2. In l.101, does the “task-relevant stimulus” refer only to the target or both the target and auditory cue? Is the auditory cue be heard from the spatially center, left, or right side of the screen? It would be better to explicitly state whether the auditory cue guides exogenous attention (e.g., a sound physically presented from the left or right or endogenous attention (e.g., a centrally presented semantic stimulus) to avoid misunderstanding by the reader.

3. Is VC the motion of the face with the lips uttering the word indicating “left” or “right” in the center of the screen as a visual cue (VC)? If my understanding is correct, this seems to be a subtle stimulus that may or may not work as a “cue.” If the participants cannot instantly understand the meaning of the VC, it will not work on its own to guide attention. Is there any data that investigates this point? At least, it is necessary to describe from the subjective point of view whether the VC is a stimulus that can be understood immediately. From the description after the results, it appears that VCs do not work by themselves, but it would be easier to understand if they were mentioned in the introduction or in the methods section.

4. The experimental procedure section should contain details about the stimuli to be presented. For example, how many centimeters away from the participant were the images presented, and how large were they? Also, how many visual angles between the center and the cues or targets that appeared? Without these details, the reader may be misled. In my case, I first thought that the positions of the targets presented on the left and right sides were fixed, but later I realized that there was some variation in the Results section. Note that, as mentioned in point 2, the location of the source of the auditory stimuli should also be described.

5. A more systematic arrangement of figures and tables throughout the Results section might be needed.

6. The Authors should provide effect sizes for statistical tests. Also, it would be easier to understand the data if variability information is added to all the graphs.

7. In addition to the Stroop effect and the Simon effect, there is another effect that could interfere with the effect of cueing in this study: the foreperiod effect. It has been widely pointed out in the research field of the cognitive function called temporal preparation that, in an experimental situation where the target is presented after different foreperiods, the degree of readiness for the target is low at the beginning of the trial, but increases over time. The results of this study might be explained in terms of the combined effect of temporal preparation and the difference in the time it takes to process each cue. If possible, I would like to see some analysis or discussion that can shed light on this perspective. This interpretation seems to be more valid than citing examples where spatial attention is efficiently guided by the integration of multiple unisensory spatial cues which are valid on themselves. This is because neither the auditory nor visual unisensory cues did not have a major effect on the efficiency of target discrimination on their own.

6. PLOS authors have the option to publish the peer review history of their article (what does this mean?). If published, this will include your full peer review and any attached files.

Reviewer #1: No

---

## [Author Response · Author response to Decision Letter 0]

21 Apr 2022

Please see Response to Reviewers document.

---

## [Decision Letter · Decision Letter 1]

26 May 2022

PONE-D-21-38703R1Enhancing performance with multisensory cues in a realistic target discrimination taskPLOS ONE

Dear Dr. Cinel,

Thank you for submitting your manuscript to PLOS ONE. After careful consideration, we feel that it has merit but does not fully meet PLOS ONE’s publication criteria as it currently stands. Therefore, we invite you to submit a revised version of the manuscript that addresses the points raised during the review process.

 Editor comments. The referee commented on your work and found most of the issues properly addressed, despite some remaining points. I also read the paper myself, and in agreement with the referee, I think it is close to the acceptance stage. I would ask you to address the remaining points before the manuscript will be officially accepted. This means, there will be no further reviewer round and the final revision serves only to give the manuscript the proper fine-tuning.

We look forward to receiving your revised manuscript.

Kind regards,

Michael B. Steinborn, PhD

Section Editor

PLOS ONE

Journal Requirements:

Reviewers' comments:

Reviewer's Responses to Questions

**Comments to the Author**

1. If the authors have adequately addressed your comments raised in a previous round of review and you feel that this manuscript is now acceptable for publication, you may indicate that here to bypass the “Comments to the Author” section, enter your conflict of interest statement in the “Confidential to Editor” section, and submit your "Accept" recommendation.

Reviewer #1: All comments have been addressed

2. Is the manuscript technically sound, and do the data support the conclusions?

Reviewer #1: Yes

3. Has the statistical analysis been performed appropriately and rigorously? 

Reviewer #1: No

4. Have the authors made all data underlying the findings in their manuscript fully available?

Reviewer #1: Yes

5. Is the manuscript presented in an intelligible fashion and written in standard English?

Reviewer #1: Yes

6. Review Comments to the Author

Reviewer #1: Thanks to the Authors for responding to my comments. The revised manuscript addressed my points adequately. I now understand that the authors' purpose is to provide a practical guideline, not to reveal multisensory cueing mechanisms theoretically.

I have two final points that should be addressed:

I think the difficulty in discussing the results is derived from the fact that each cue may have a different length of interpretation time (see lines 433-436 in the marked-up version of the manuscript). We agree that the temporal preparedness with the definitions used by the authors might not affect the results. On the other hand, I think it is worth noting that different interpretation times for each cue affected the results, as described in lines 603-612 in the marked-up version of the manuscript. In such a case, I do not think that the results comparing the effect of cue type on performance within each SOA would have allowed the authors to compare the effect of cues eliminating differences in SOA in terms of the psychological factors. Although the authors appeared not to explicitly state that this analysis was to remove the effect of SOA theoretically, readers might interpret it that way. Suppose the authors note that examining the effects of cues within each SOA is only meant to provide a practical guideline for the authors' purposes, not to remove the effects of SOA theoretically. In that case, the use of these statistical analyses might be acceptable.

The error bars of different colors in Figures 6 and 7 are covered, and some are not visible. Therefore, it is recommended that the error bars be separately presented, as shown in Figures 2 and 3.

7. PLOS authors have the option to publish the peer review history of their article (what does this mean?). If published, this will include your full peer review and any attached files.

Reviewer #1: No

---

## [Editor Report · Decision Letter 2]

18 Jul 2022

Enhancing performance with multisensory cues in a realistic target discrimination task

PONE-D-21-38703R2

Dear Dr. Cinel,

We’re pleased to inform you that your manuscript has been judged scientifically suitable for publication and will be formally accepted for publication once it meets all outstanding technical requirements.

Kind regards,

Michael B. Steinborn, PhD

Section Editor

PLOS ONE
---

## [Editor Report · Acceptance letter]

28 Jul 2022

PONE-D-21-38703R2 

Enhancing performance with multisensory cues in a realistic target discrimination task 

Dear Dr. Cinel:

I'm pleased to inform you that your manuscript has been deemed suitable for publication in PLOS ONE. Congratulations! Your manuscript is now with our production department. 

Kind regards, 

on behalf of

Dr. Michael B. Steinborn 

Section Editor

PLOS ONE